# Resistomycin Induced Apoptosis and Cycle Arrest in Human Hepatocellular Carcinoma Cells by Activating p38 MAPK Pathway In Vitro and In Vivo

**DOI:** 10.3390/ph14100958

**Published:** 2021-09-23

**Authors:** Zhuo Han, Xingming Zhao, E Zhang, Jiahui Ma, Hao Zhang, Jianjiang Li, Weidong Xie, Xia Li

**Affiliations:** 1Marine College, Shandong University, Weihai 264209, China; hanzhuo1013@gmail.com (Z.H.); zhaoxingming@mail.sdu.edu.cn (X.Z.); 202020923@mail.sdu.edu.cn (E.Z.); sdumjh@gmail.com (J.M.); zh518294@gmail.com (H.Z.); 202017645@mail.sdu.edu.cn (J.L.); wdxie@sdu.edu.cn (W.X.); 2School of Pharmaceutical Sciences, Shandong University, Jinan 250012, China

**Keywords:** resistomycin, apoptosis, cell cycle arrest, hepatocellular carcinoma, p38 MAPK

## Abstract

Resistomycin, a quinone-related natural antibiotic, has shown strong inhibitory activity against human hepatocellular carcinoma (HCC) in vitro. Here, we investigated the role of p38 MAPK in the pro-apoptotic and G2/M phase arrest action of HCC HepG2 cells upon treatment with resistomycin in vitro and in vivo. Our results showed that resistomycin dose- and time-dependently reduced the viability of HepG2 cells and also showed lower cytotoxicity in normal human kidney cells (293T) and hepatocyte cells (HL-7702). Resistomycin treatment induced apoptosis and cell cycle arrest in HepG2 cells, accompanied by changes in the expression of related proteins, including Bax, Cyclin B1, etc. Surprisingly, resistomycin-mediated apoptotic cell death and cell cycle arrest were impeded by SB203580 (an inhibitor of p38 catalytic activity), suggesting that p38 MAPK signaling may play an important role that impedes eventual cell death. In this connection, data in vitro and in vivo demonstrated that resistomycin increased the phosphorylation of p38 and MAPKAPK-2 in HepG2 cells. Furthermore, we provided evidence that p38 signaling is involved in resistomycin-induced p38 MAPK pathway effects in HCC, using computer docking models. Our study indicated that resistomycin activates the p38 MAPK signaling pathway by which the growth of HepG2 cells is suppressed for apoptosis and G2/M phase arrest in vitro and in vivo, and it is a promising therapeutic leading compound for drug development in HCC treatment.

## 1. Introduction

According to the latest global cancer burden data from the International Agency for Research on Cancer in 2020, the number of new live cancers increased by about 900,000 (4.7%), ranking sixth in new cancer cases, and the more than 800,000 (8.3%) deaths ranked third [1]. The third primary cause of cancer death is liver cancer and hepatocellular carcinoma (HCC) plays a significant role in this [1,2,3]. The main etiological factors of HCC include chronic hepatitis (B or C) virus infection and alcoholic liver disease [4,5]. Despite many advanced therapies for HCC being used over the past few decades, the total cure rate remains low due to the high incidence of metastasis, recurrence and easy drug resistance [6]. Sorafenib was the first drug approved by the Food and Drug Administration (FDA) in 2005 to treat advanced hepatic cancers and, soon afterwards, the China Food and Drug Administration (CFDA) also approved use of the drug [7,8]. About 10 years later, regorafenib, the second drug for HCC treatment, was approved by the FDA and CFDA [9]. Hence, further new promising therapies are needed for patients with HCC.

Mitogen-activated protein kinase (MAPK) pathways are the extensively studied molecular pathways. While initial investigations were more about defining the components and organization of MAPK pathways, recent research has focused on the physiological functions of these kinases [10,11]. p38 MAPK, one of the subfamilies of MAPKs, has been confirmed to be associated with cell growth, inflammatory responses and also cell cycle progression [12,13]. However, the functions of p38 MAPK activation vary between specific cell types. A study by Iyoda and colleagues revealed that the activities of p38 MAPK and its upstream MKK6 in HCC were significantly lower than in adjacent non-neoplastic liver tissue, suggesting that the reduction in p38 MAPK activities provides a growth advantage to human HCC, in part by resisting apoptosis [14]. In relation to this, exploring novel anti-cancer agents that activate the p38 MAPK pathway may be of benefit in the treatment of HCC.

Resistomycin, a unique quinone-related natural antibiotic, exhibits broad-spectrum antibacterial and vasoconstrictive activity. It suppresses RNA and protein synthesis without any inhibitory effect on DNA synthesis [15]. In addition, it has been suggested that resistomycin can be used as an anti-cancer compound because it exhibits strong cytotoxicity against HCC, cervical carcinoma cell lines and gastric cancer cells in vitro [16,17].

In the latest research, resistomycin, as an inhibitor of Pellino-1(ubiquitin E3 ligase), inhibited invasion and metastasis in triple-negative breast cancer. Here, we report that resistomycin induced apoptosis and G2/M phase arrest of HepG2 cells in vitro and in vivo, with the p38 MAPK signaling pathway being involved in this action. These results may shed light on a new strategy for HCC therapy.

## 2. Materials and Methods

### 2.1. Reagents and Cell Lines

Resistomycin (>98%) was dissolved and diluted in 1 mmol/L stock solutions with dimethyl sulfoxide (DMSO) and with DMSO as a replacement for resistomycin in the control group. Antibodies specific for Bcl-2, Bax, GAPDH Cdc2, Cdc25A, Cyclin B1, phosphor-Cdc2 (Tyr15), phosphor-p38 (Thr180), phosphor-MAPKAPK-2 (Thr222) and p38 MAPK were purchased from Cell Signaling Technology (CST, Inc., Beverly, MA, USA). Human HCC cell lines, including HepG2, PLC-PRF-5, SMMC-7721 and normal human kidney cell line 293T were purchased from the Shanghai Institute for Biological Sciences (SIBS, Shanghai, China) and cultured as requested. Normal human hepatocyte cell line HL-7702 was purchased from BeNa Culture Collection (Kunshan, Jiangsu, China) with the same culture.

### 2.2. Cell Cytotoxicity Assay

The cytotoxicity effect of resistomycin on cancer cells and normal cells was assessed by MTT (3-(4,5-dimethylthiazol)-2,5-diphenyltetrazolium bromide) assay (Sigma-Aldrich Corp., St. Louis, MO, USA). Cells were treated with various concentrations of resistomycin in the absence or presence of 2.5 or 5 μmol/L SB203580 (Beyotime Institute of Biotechnology, Shanghai, China) pretreatment for 1 h. The negative control was treated with the same volumes of DMSO. MTT was added to each well of the plate after 24–72 h of continuous culture. About 3.5–4 h later the medium was sucked away and 150 μL of DMSO was added to each well. Then, the absorbance in the plate was measured at 570 nm and the half-maximal inhibitory concentration (IC_50_) values were calculated from the plotted results, using the negative control as 100% cell survival.

### 2.3. DAPI, JC-1 and Annexin V/PI Staining

The change in nucleus and the fluorescence intensity were observed by means of DAPI, JC-1 and Annexin V/PI (Beyotime Institute of Biotechnology, Shanghai, China) staining assay after resistomycin treatment for 24 h, which we conducted as previously described [18].

### 2.4. Flow Cytometry Analysis of Apoptosis

Cells were treated with the indicated concentration of resistomycin for 24 h in the absence or presence of 5 μmol/L SB203580 pretreatment, as mentioned above. The rate of cell apoptosis can be detected using an Annexin V-FITC apoptosis kit, as mentioned previously [19].

### 2.5. Cell Cycle Distribution

Cells stained with PI (propidium iodide; Beyotime Institute of Biotechnology, Shanghai, China) were detected using flow cytometry (Becton Dickinson FACScan, San Jose, CA, USA) to measure cell cycle distribution after resistomycin treatment for 24 h (with or without 5 μmol/L SB203580 pretreatment) according to our previous description [18].

### 2.6. Western Blotting Tumor Transplantation and Administration

SB203580 (5 μmol/L) was added (or not) before cells were incubated with different concentrations of resistomycin for 24 h. Exacted proteins were measured by western blotting, which we performed as previously described [19].

### 2.7. Tumor Transplantation and Administration

HepG2 cells were digested, centrifuged and resuspended. Next, the same volume of Matrigel (Corning, Inc, NY, USA) was added to the cell suspension to achieve a density of 10^7^/ml cells. Then, 100 μL of cells was inoculated into the nude mice (Hangzhou Ziyuan Animal Technology Co., Ltd., Hangzhou, China (Certificate No. SCXK(Zhe) 2019-0004)) arm to establish transplanted models. After the volumes of tumors grew to 100–200 mm^3^, the mice were randomly divided into four groups (*n* = 6): control groups; low-dose groups of 10 mg/kg resistomycin; high-dose groups of 20 mg/kg resistomycin; positive groups of 10 mg/kg 5-fluorouracil (5-Fu).

Six-week-old Balb/c athymic (nu+/nu+) male mice (Hangzhou Ziyuan Animal Technology Co., Ltd., Hangzhou, China (Certificate No. SCXK(Zhe) 2019-0004)) were given 100 μL of 10 or 20 mg/kg resistomycin with gastrin infusion and 100 μL of 10 mg/kg 5-Fu by intraperitoneal injection 2 days at a time. At the same time, the volumes and weights of mice were measured. The tumors were peeled off, weighed and stored at −80 °C, and the blood was collected to blood collection tubes once the mice were killed after treatment for 15 days. Then blood was left to set for 30 min at room temperature and was extracted by centrifugal force for 10 min at 3000 rpm. All animal experiments followed the institutional guidelines and rules formulated by the Animal Care and Use Committee at Shandong University. Mice were acclimatized to the laboratory room and handling for 2 weeks before the start of the experiments.

### 2.8. Tumor Protein Extraction

Tumors dissected into pieces weighing about 100 mg were cleaved in 400 μL of lysis buffer, broken up further by homogenization and then centrifuged twice at 12,000 rpm for 30 min at 4 °C. Protein quantitation and western blot assay were performed as described above.

### 2.9. Docking Studies

Molecular docking research was performed using the related software, AutoDockTools-1.5.6. The p38 3D structure (PDB ID: 6SFI, 1.6 A resolution) was obtained in the PDB. In docking analysis, the conformation that has the lowest binding energy is selected as the most suitable for drugs and ligands. The Lamarck genetic algorithm is used as a docking method for docking analysis.

### 2.10. Statistical Analysis

Data of triplicate independent experiments were presented as the mean ± standard deviation. Tukey’s multiple comparison test of one-way analysis of variance (ANOVA) in GraphPad Prism 8.01 was used for evaluating the data. Significant differences are presented as: * *p* < 0.05; ** *p* < 0.01; *** *p* < 0.001.

## 3. Results

### 3.1. Resistomycin Reduces Cell Viability in Human Liver Cancer and Normal Cell Lines

MTT analysis was first carried out as a cytotoxicity test of resistomycin (Figure 1A) on several subtypes of HCC cells (HepG2, SMMC-7721 and PLC-PRF-5), normal human kidney cells (293T) and hepatocyte cells (HL-7702). Resistomycin exhibited the strongest sensitivity on HepG2 cells, with lower cytotoxic effects on normal human kidney cells (293T) and hepatocyte cells (HL-7702). The IC_50_ values indicate that resistomycin inhibits 50% of cell activity after 48 h of treatment for HepG2, SMMC-7721, PLC-PRF-5, HL-7702, Huh7 and 293T, with values of 0.25 ± 0.02, 0.46 ± 0.06, 1.10 ± 0.14, 1.13 ± 0.39, 0.35 ± 0.21 and 3.06 ± 0.30 μmol/L, respectively (Table 1). In addition, resistomycin exerted its anti-proliferative effect in a dose- and time-dependent manner (Figure 1B). The IC_50_ values for resistomycin treatment of HepG2 cells for 24, 48 and 72 h were 1.31 ± 0.15, 0.25 ± 0.02 and 0.059 ± 0.002 μmol/L, respectively. Hence, we used HepG2 cells to further study the anti-cancer effect and mechanism of action of resistomycin.

### 3.2. Resistomycin Induces Apoptosis in HepG2 Cells

To characterize the potential mechanism responsible for the anti-cancer activity of resistomycin, we observed the morphology of HepG2 cells. First, a correlative pharmacodynamic experimental study was conducted for qualitative analysis (Figure 1C). DAPI (4′,6-diamidino-2-phenylindole) staining revealed that resistomycin treatment for 24 h caused characteristic apoptotic morphological alterations in a dose-dependent manner, including nuclear condensation and apoptotic body formation, and the fluorescence intensity increased. JC-1 was used as a probe to detect the mitochondrial membrane potential (MMP). When cells presented apoptotic changes, the MMP decreased and aggregates transformed into singletons. JC-1 staining showed that the intensity of green fluorescence was enhanced in JC-1 monomer compared to a lessening of red fluorescence in JC-1 polymer after the cells were treated with different concentrations of resistomycin for 24 h. For quantitative analysis, JC-1 can be adopted to calculate the apoptotic index by flow cytometry. The ratio of red to green fluorescent intensity decreased significantly in 0–0.5 μmol/L resistomycin for 24 h, which was 1.04, 0.59, 0.23 and 0.13, respectively (Figure 1D,E).

During apoptosis, phosphatidylserine redistributes from the inter leaflet of the plasma membrane to the outer layer, which can be bound with Annexin V and detected through staining and flow cytometry. The strength of FITC (fluorescein isothiocyanate) fluorescence gradually increased in the treated groups relative to no fluoresce in the control group (Figure 2A). Likewise, the pro-apoptotic function of resistomycin can be calculated by co-staining with Annexin V-FITC and PI. The ratios of Annexin V-positive cells remarkably aggrandized, as detected by flow cytometer after resistomycin effects on HepG2 cells. The percentages of Annexin V-positive cells were 4.52%, 15.98%, 34.54% and 37.17%, respectively (Figure 2B,C). Along with cell apoptosis, Bax (a pro-apoptotic protein) increased but Bcl-2 (an anti-apoptotic protein) decreased in HepG2 cells (Figure 2D,E). Therefore, we suggest that resistomycin exerts its anti-cancer activity by inducing apoptosis in HepG2 cells.

### 3.3. Resistomycin Induces G2/M Phase Arrest in HepG2 Cells

To investigate the mechanism of resistomycin in detail, we examined the cell cycle process affected by the drug through flow cytometry analysis. Resistomycin effectually caused an accumulation of cells in the G2/M phase. The percentages of cells in the G2/M phase treated with resistomycin were 9.36%, 16.61%, 25.34% and 37.56%, respectively (Figure 3A,B).

The Cdc2–Cyclin B1 complex controls the interim G2/M period in a normal cell cycle and activation of the complex at the G2/M boundary can be mediated by Cdc25A through dephosphorylating Cdc2 at Tyr15 [20]. For this reason, we performed western blotting to analyze the changes in expression levels of these regulators after the cells were treated with resistomycin. In response to resistomycin treatment, expressions of Cdc2, Cdc25A and Cyclin B1 were downregulated, whereas phosphor-Cdc2 and phosphor-Cdc25A were upregulated in HepG2 cells, which confirmed that HepG2 cells were impeded in the G2 period before entering the mitotic phase after being exposed to different concentrations of resistomycin (Figure 3C,D).

### 3.4. Resistomycin Induces Apoptosis and G2/M Phase Arrest by Activating the p38 MAPK Pathway

We also used western blotting to evaluate the expression of several proteins in order to gain a better understanding of the underlying mechanism of resistomycin-induced apoptosis and G2/M phase arrest in HepG2 cells. Surprisingly, resistomycin treatment dose-dependently upregulated the expression of phosphor-p38 and phosphor-MAPKAPK-2 but expression of p38 remained unchanged (Figure 4A,B), suggesting the possibility that p38 MAPK may be associated with the anti-cancer activities of resistomycin.

SB203580, a cytokine-suppressive anti-inflammatory drug, is widely used in delineating the function of p38 MAPK, with no inhibitory effect on ERK [21]. SB203580 (5 μmol/L) was added to cells in advance to assess if p38 MAPK was involved with the resistomycin-induced apoptosis and G2/M phase arrest. The inhibition of p38 MAPK resulted in decreases in apoptotic cells from 23.15% to 9.80% and from 34.95% to 26.28% with 0.25 and 0.5 μmol/L resistomycin treatment, respectively (Figure 4C,D). Moreover, western blotting showed that after dealing with 5 μmol/L SB203580, the expression of Bcl-2 was increased and the expression of Bax, p38, phosphor-p38 and phosphor-MAPKAPK-2 was decreased in comparison with resistomycin treatment only (Figure 4E).

Additionally, the inhibition of p38 MAPK obviously downregulates the ability of resistomycin to induce G2/M phase arrest. The proportion of cells in the G2/M phase reduced from 25.39% to 19.22% and from 32.31% to 26.52% when treated with 0.25 and 0.5 μmol/L resistomycin, respectively (Figure 5A,B). Consistent with flow cytometry data, after dealing with 5 μmol/L SB203580, the expression of Cdc2, Cyclin B1 and Cdc25A was increased and the expression of phosphor-Cdc2 and phosphor-Cdc25A was decreased in comparison with resistomycin treatment only (Figure 5C). These findings confirmed the involvement of the p38 MAPK signaling pathway in resistomycin-induced apoptosis and G2/M phase arrest.

### 3.5. Resistomycin Suppressed Human Hepatocyte HepG2 Cell Growth in Xenograft Models

The HepG2 cells were transplanted into nude mice in order to observe the inhibition effects of resistomycin in vivo. The mice were given 10 and 20 mg/kg of resistomycin by gavage and 10 mg/kg 5-fluorouracil (5-Fu) as a positive drug for HCC treatment [22] by intraperitoneal injection every 2 days. Weights and volumes were measured on the same day. After 15 days, tumors were peeled off and weighed once the mice were killed. The results revealed that the volume growth tendency of hepatoma increased more slowly in the treated compared with the control group. The tumor weights also indicated the same effect of tumor inhibition (Figure 6A–C).

### 3.6. Resistomycin Restrained Apoptosis and G2/M Phase Arrest by Phosphorylating p38 in the Mice Model

The proteins of tumor tissue extraction and western blot assay further showed that resistomycin induced apoptosis and cell cycle arrest by activating the p38 MAPK pathway. Protein expression of Bcl-2, Cyclin B1 and Cdc25A was upregulated, whereas the expression of Bax, p-Cdc2, p-Cdc25A, p-p38 and p-MAPKAPK-2 was downregulated. This further verified the in vivo viewpoint that resistomycin activated p38, leading to apoptosis and G2/M phase arrest of HepG2 cells (Figure 6D).

### 3.7. Molecular Docking Analysis of Resistomycin Activate p38

In order to further determine whether resistomycin activates the p38 MAPK pathway by interacting with p38, as well as to predict its binding mode at the active site of p38, we employed the AutoDockTools-1.5.6 software package to conduct a molecular docking simulation study. During software simulation of molecular docking, the conformation that has the lowest binding energy is selected as the most suitable binding site for drugs and ligands. According to the simulated docking results, the optimal binding mode of resistomycin and p38 is shown in Figure 7. The results of the docking analysis showed that the lowest binding energy of resistomycin was −6.92 kcal/mol and the inhibition constant (Ki) was 8.52 μM. Additionally, we found that resistomycin is in a favorable position in the p38 domain and binds through hydrogen-bond interaction.

## 4. Discussion

Resistomycin is a quinone-related natural antibiotic that exhibits broad-spectrum antibacterial activity. Studies have reported that resistomycin may serve as an anti-cancer compound with strong cytotoxicity against several human carcinoma cell lines in vitro [16]. The anti-tumor effect of resistomycin has also been put down to mitochondrial dysfunction and its direct binding to E3 ligase Pellino-1 [23,24,25]. In the present study, we demonstrated that resistomycin treatment significantly induced strong anti-cancer activity in HepG2 cells, with lower cytotoxicity in normal human hepatocyte HL-7702 cells and kidney 293T cells, which may be a therapeutic window for resistomycin. Immunofluorescence images displayed the characteristic features and fluorescence intensity changes of resistomycin-induced apoptotic cells by immunofluorescence microscopy. Moreover, the phenomenon of phosphatidylserine externalization treated with resistomycin was more evident, as presented quantitatively by flow cytometry. Furthermore, we found that resistomycin upregulated Bax (a pro-apoptotic protein) but downregulated Bcl-2 (an anti-apoptotic protein) expression by western blotting evaluation. These results showed that resistomycin may trigger apoptosis in HCC cells.

Cell cycle interference is one of the most important mechanisms implicated in the cytotoxic effects and apoptosis of anti-cancer drugs. In this paper, we also confirmed that resistomycin treatment led to G2/M phase arrest in HepG2 cells. We also observed that resistomycin induced G2/M phase retardation and apoptosis in HepG2 cells by activating the p38 MAPK signaling pathway in vitro and in vivo, and that inhibition of p38 MAPK weakened the effects of resistomycin in cell cycle retardation and apoptosis.

It is well known that activation of p38 MAPK plays an important role in apoptosis and cell cycle retardation when it receives various extracellular stimuli in human cancer [26]. For instance, p38 MAPK activation results in apoptosis and G2/M retardation induced by oxidative stimuli in HepG2 cells, whereas SB203580 pretreatment reversed this action in the same cells [27]. p38 MAPK is relatively inactive in its unphosphorylated form and is activated by phosphorylation of threonine and tyrosine residues [28]. The activation of p38 MAPK by phosphorylation leads to the activation of transcription factors, including MAPKAPK-2, MAPKAPK-3, p53 and p73 [29,30,31]. Downstream activation contributes to multiple responses, such as cell cycle arrest, autophagy, apoptosis and cell differentiation [32], suggesting that p38 MAPK is a potential target for anti-cancer therapies. Here, we noticed that the expression levels of phosphor-p38 and phosphor-MAPKAPK-2 significantly increased in HepG2 cells after resistomycin treatment. Pretreatment with SB203580, a specific inhibitor of p38 MAPK, functionally decreased the expression levels of phosphor-p38 and phosphor-MAPKAPK-2. Moreover, pretreatment with SB203580 effectively reversed the function of resistomycin on apoptosis and cell cycle arrest, thereby regulating the cytotoxicity of resistomycin. Furthermore, we established a xenotransplantation model to confirm that resistomycin not only inhibited the growth of tumors but also induced apoptosis and G2/M phase retardation in vivo. In addition, an interaction between resistomycin and p38 was found through computer simulations of molecular docking, strongly supporting the notion that resistomycin activates the p38 MAPK pathway through interacting with p38.

In this article, we firstly found resistomycin could induce G2/M phase arrest in HepG2 cells through the p38 MAPK pathway. Although resistomycin had underlying anti-cancer activity, with IC50 of 0.25 µmol/L against HepG2 cells and 1.13–3.06 μmol/L against non-cancerous liver and kidney mammary cells, developing it into HCC therapeutics by inhibition of the p38 MAPK pathway needs to be cautiously performed. Toxicity experiments in vivo might warrant further investigation. In addition, a more detailed exploration of the mechanism responsible for activation of the p38 MAPK signaling pathway and how this activation influences cell apoptosis and G2/M phase arrest was needed.

## 5. Conclusions

Taken together, our results verified that activation of the p38 MAPK signaling pathway is involved in resistomycin-induced apoptosis and G2/M phase arrest in HepG2 cells, and that the inhibition of p38 MAPK activity lowers the efficiency of resistomycin treatment. This indicates that resistomycin treatment may be considered as a potential strategy for HCC treatment.

## Figures and Tables

**Figure 1 pharmaceuticals-14-00958-f001:**
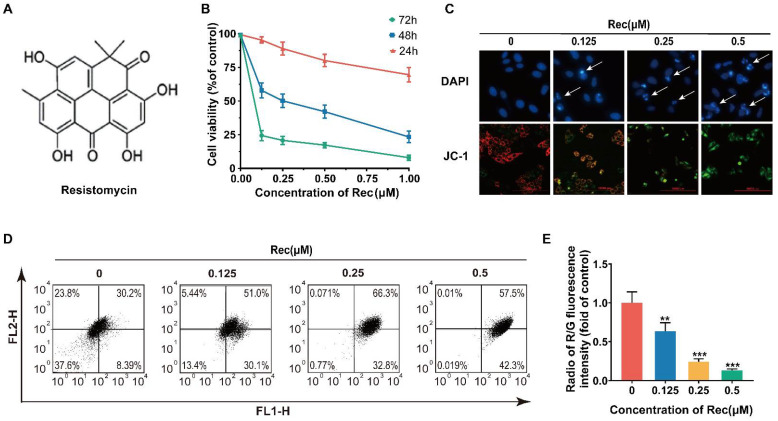
Resistomycin suppressed proliferation and decreased mitochondrial membrane potential (MMP) of HepG2 cells. (**A**) Structure of resistomycin. (**B**) Resistomycin exerted its anti-proliferative effect in a dose- and time-dependent manner when HepG2 cells were treated with resistomycin for 24 h, 48 h and 72 h and cell viability was determined using the MTT assay. (**C**) The results of DAPI and JC-1 staining after HepG2 cells were treated with different concentrations of resistomycin in 24 h. Arrow: apoptotic body in DAPI staining. JC-1: red fluorescence represents high mitochondrial membrane potential forming aggregates, green fluorescence represents low mitochondrial membrane potential forming monomers. Scale bar: 10000 μm (DAPI and JC-1). (**D**,**E**) Assay of flow cytometry showing the proportion of FL2-H to FL1-H in HepG2 cells treated with 0–0.5 μM resistomycin for 24 h. ** *p* < 0.01 and *** *p* < 0.001 versus 0 μM resistomycin.

**Figure 2 pharmaceuticals-14-00958-f002:**
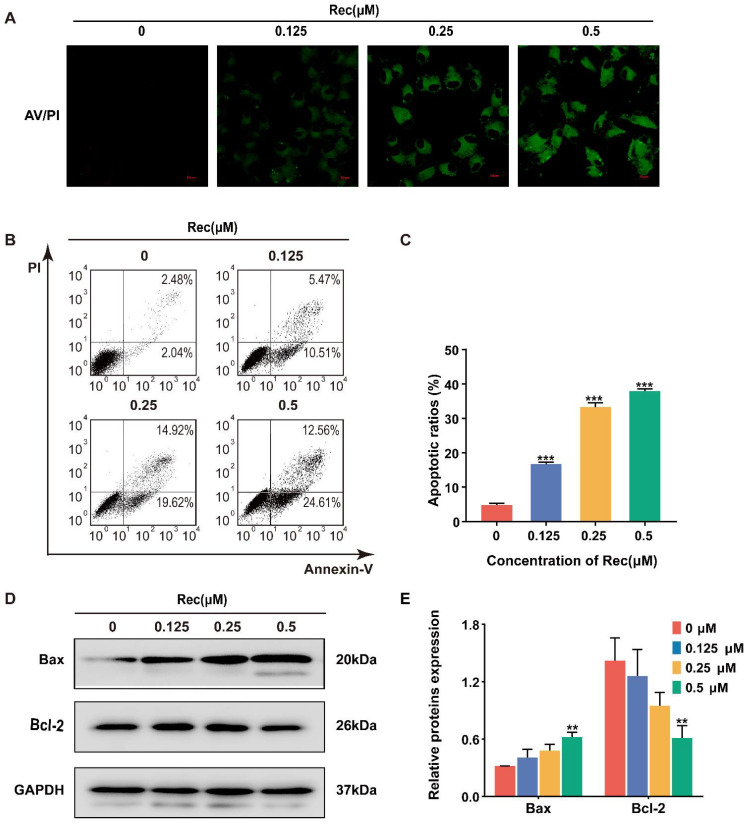
Resistomycin induced apoptosis in HepG2 cells. (**A**) The results of Annexin V/PI staining after HepG2 cells were treated with different concentrations of resistomycin in 24 h. Green fluorescence represents apoptotic cells. Scale bar: 20 μm. (**B**,**C**) Number and percentage of apoptotic cells measured by flow cytometry with 0, 0.125, 0.25 and 0.5 μM resistomycin for 24 h. (**D**,**E**) Western blot analysis showing decreased pro-apoptotic protein Bax and increased anti-apoptotic protein Bcl-2. ** *p* < 0.01 and *** *p* < 0.001 versus 0 μM resistomycin.

**Figure 3 pharmaceuticals-14-00958-f003:**
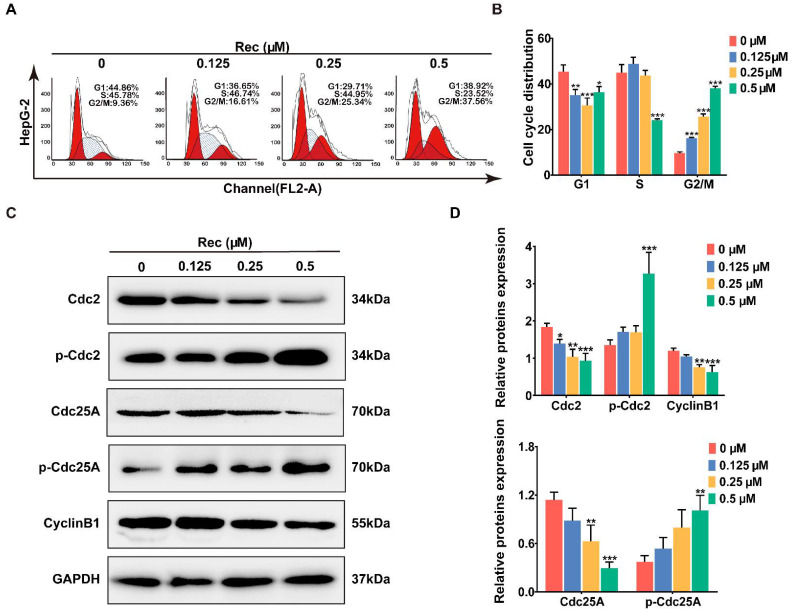
Resistomycin caused G2/M phase arrest in HepG2 cells. (**A**,**B**) Flow cytometry analysis of the distribution of HepG2 cells in the G1, S and G2 phases after treatment with 0, 0.125, 0.25 and 0.5 μM resistomycin for 24 h (each phase presented as G1-S-G2/M: red-stripes-red). (**C**,**D**) Western blot analysis showing that resistomycin downregulated the expression of Cdc25A, Cdc2 and Cyclin B1 and upregulated the expression of p-Cdc2 and p-Cdc25A after treatment with 0, 0.125, 0.25 and 0.5 μM resistomycin for 24 h. * *p* < 0.05, ** *p* < 0.01 and *** *p* < 0.001 versus 0 μM resistomycin.

**Figure 4 pharmaceuticals-14-00958-f004:**
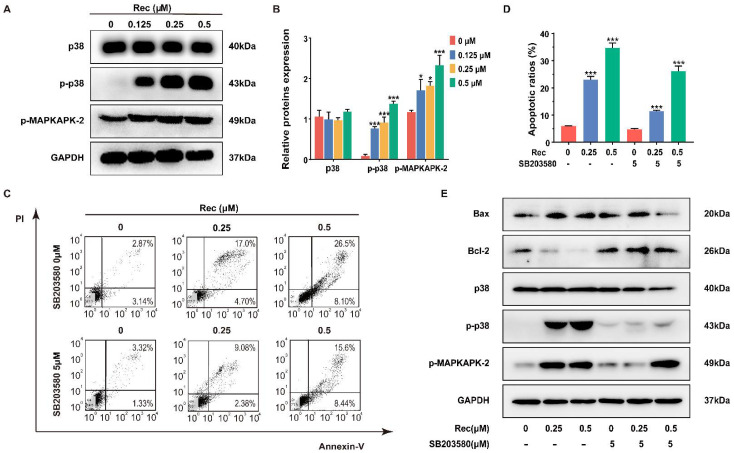
Resistomycin induced apoptosis by activating p38 MAPK signaling in vitro, which was impeded by using SB203580, an inhibitor of p38 MAPK. (**A**,**B**) The expression changes in the p38 MAPK pathway in HepG2 cells were triggered by resistomycin treatment for 24 h and relative protein expressions were detected via western blotting. (**C**,**D**) Flow cytometry analysis of the numbers and percentage of apoptotic cells treated with resistomycin for 24 h after 5 μmol/L SB203580 was added to the HepG2 cells. (**E**) HepG2 cells pre-treated with or without 5 μM SB203580 for 1 h and exposed to various concentrations of resistomycin for another 24 h, then analyzed by western blotting to measure the alteration of relative protein expressions. * *p* < 0.05 and *** *p* < 0.001 versus 0 μM resistomycin.

**Figure 5 pharmaceuticals-14-00958-f005:**
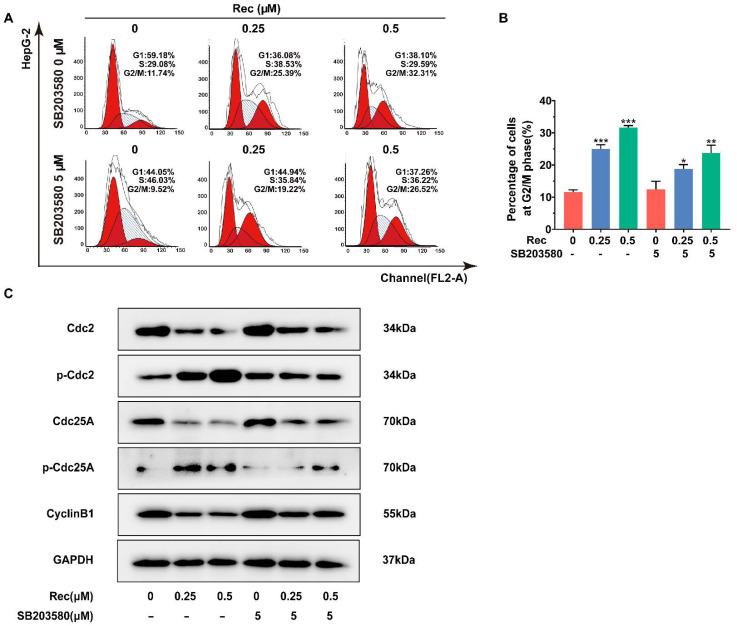
The p38 MAPK inhibitor SB203580 suppressed cycle arrest induced by resistomycin in HepG2 cells. (**A**,**B**) The ratios of G2/M phase cells treated with resistomycin (with or without 5 μM SB203580) were detected by flow cytometry (each phase presented as G1-S-G2/M: red-stripes-red). (**C**) HepG2 cells pre-treated with or without 5 μM SB203580 for 1 h and exposed to various concentrations of resistomycin for another 24 h, then analyzed by western blotting to measure the alteration of proteins related to cell cycle arrest. * *p* < 0.05, ** *p* < 0.01 and *** *p* < 0.001 versus 0 μM resistomycin.

**Figure 6 pharmaceuticals-14-00958-f006:**
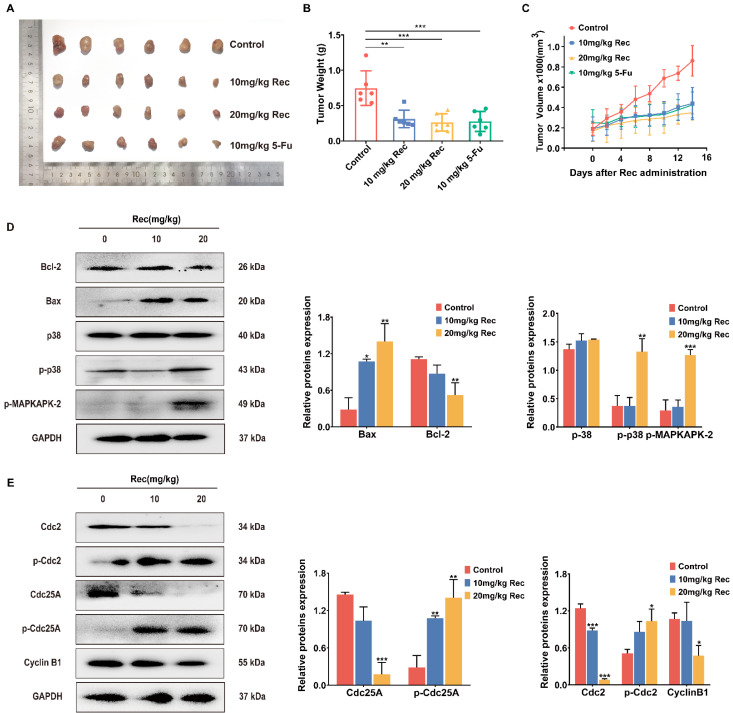
Resistomycin resulted in apoptosis and G2/M phase arrest by activating p38 MAPK signaling in vivo. (**A**) Images of transplanted HepG2 tumors treated with solvent, resistomycin (10 and 20 mg/kg) and 5-Fu (10 mg/kg). (**B**) Tumor weight changes with treatment. (**C**) Curve of tumors indicating that resistomycin effectively controls the growth of xenograft tumors. (**D**) Relative protein expression related to apoptosis, cell cycle arrest and the p38 MAPK pathway in HepG2 tumors. * *p* < 0.05, ** *p* < 0.01 and *** *p* < 0.001 versus 0 mg/kg resistomycin.

**Figure 7 pharmaceuticals-14-00958-f007:**
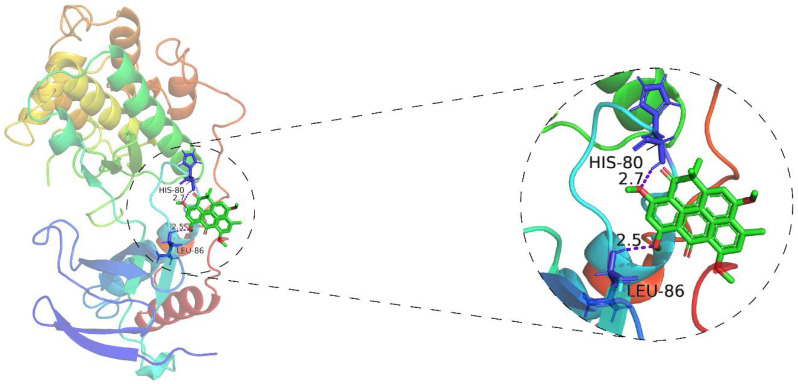
AutoDockTools software was employed to obtain the optimal docking model of resistomycin to the p38, and the resulting image is a panoramic view of the optimal binding method (Purple dotted line: hydrogen-bond). The lowest binding energy configuration is −6.92 kcal/mol.

**Table 1 pharmaceuticals-14-00958-t001:** The IC50 values for resistomycin cytotoxicity after 48 h on human liver cancer cell lines and normal cell lines.

Cell line	IC50 (μmol/L)	
	Resistomycin	Doxorubicin
HepG2	0.25 ± 0.02	0.84 ± 0.05
SMMC-7721	0.46 ± 0.06	1.37 ± 0.08
PLC-PRF-5	1.10 ± 0.14	1.33 ± 0.10
HL-7702	1.13 ± 0.39	1.33 ± 0.02
Huh7	0.35 ± 0.21	1.34 ± 0.05
293T	3.06 ± 0.30	1.54 ± 0.25

Cytotoxicity of resistomycin and doxorubicin on human hepatoma cells, normal human liver cells and human normal renal cells. Different concentrations of resistomycin acted on the cancer and normal cells in 48 h. The IC50 values showed in Table 1. Dates of triplicate independent experiments were handled in the method of mean ±SD.

## Data Availability

Data is contained within the article.

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
