# Peer review of "Resistomycin Induced Apoptosis and Cycle Arrest in Human Hepatocellular Carcinoma Cells by Activating p38 MAPK Pathway In Vitro and In Vivo"

_pharmaceuticals, 2021, doi:10.3390/ph14100958_

Round 1
Reviewer 1 Report
The authors have presented a paper verifying Resistomycin-induced apoptosis and cycle arrest in HCC.
The abstract is brief but very specific regarding the purpose, methods and results.
The introduction provides adequate details regarding the background issues, the aggressiveness of HCC, the involvement of MAPK and the potential role of Resistomycin.
The methods section is thorough and presents all aspects involved in the study. However, in the statistics subsection it is not mentioned whether the data was tested for normality of distribution and whether non-normal data was encountered, and if this implied changing the statistical approach.
The results are presented thoroughly and with sufficient detail, including images from Western blotting, flow cytometry, and even macroscopic images of the transplanted tumors.
The discussions focus mainly on interpreting the results in the context of other literature data. The section is rather brief but comprises essential information. Perhaps a good addition might be a short paragraph on the strengths and limitations of this study.
Some minor grammar errors throughout the text should be addressed.
Author Response
Response to Reviewer 1 Comments
Point 1: The methods section is thorough and presents all aspects involved in the study. However, in the statistics subsection it is not mentioned whether the data was tested for normality of distribution and whether non-normal data was encountered, and if this implied changing the statistical approach.

Response 1: We have described detailly in the statistics subsection of “Materials and methods” as shown in revised manuscript. We used Tukey’s multiple comparison test of one-way analysis of variance (ANOVA), the normal distribution was firstly test. Most of date conformed to Gaussian distribution. In the process of statistics, we meet a few non-normal data, but the data distributed on the both sides of the regression line in normal QQ plot and the statistical approach did not change.
Point 2: The discussions focus mainly on interpreting the results in the context of other literature data. The section is rather brief but comprises essential information. Perhaps a good addition might be a short paragraph on the strengths and limitations of this study.
Response 2: We have re-written accordingly as shown in the discussion part of revised manuscript page13,line 353 of revised manuscript.
Point 1: Some minor grammar errors throughout the text should be addressed
Response 1: We have commissioned Proof-Reading-Service. com Ltd to modified the language before submission. Now, we have checked the manuscript clearly throughout the text again. we corrected some minor grammar errors and highlighted with yellow background as shown in the revised manuscript.

Reviewer 2 Report
The authors show that Resistomycin induces apoptosis and cycle arrest in HCC cells and that p38 is involved in this process. Though the line of thought and research rational are clearly depicted, some issuses remain unresolved and should be adressed. Therefore I have several major and minor issuses that should be resolved preceeding publication.
Major issues:
- English expression and wording. I would reccommed having the English style and wording checked. I often stumbled across very uncommon expressions, which are not correct in my opinion. Also the grammar is frequently wrong.
- The figure legends in general should be revised. The legends should indicate the type of assay and treatment conditions. They should further clearly indicate what is seen (e.g. colors in images)
- Selectivity of the drug. The authors state that the drug is more effective in cancer cells, however they show that the IC50 for one of the HCC lines and the hepatocyte cell line used is almost identical (1.10 vs. 1.13). The authors fail to explain this. Further, the IC50 only differs 2-fold in the other cell lines. The authors should elaborate on why they think this drug might be selective and further cell lines should be tested. Additionally, the apoptosis markes and p38 activation should be checked (at least in key experiments) with another HCC line, as well as a healthy cell line to strenghten the claim.
- Confocal images are too small. I cannot see the effects described in the text.
- As the authors correctly pointed out that the ubiquitin ligase Pellino-1 is affected by the drug, it is hard to distinguish the contribution of p38 and Pellino-1 to the phenotype observed. The authos should try to clarify this by additional experiments (e.g. Pellino-1 inhibition or knock-down experments), or at least discuss this in more detail.
Minor issues
- line 45 "most studied" I doubt that. The context should be clarified
- line 62 "Pellino" is first mentioned here and should be explained or introduced to the reader
- Fig 1B. Normalization to 100% of Control should be visualized in the figure
- Several downstream effects of p38 activation or inhibition have been mentioned in the discussion. It would be benefitial to investigate some of these in the study.
Author Response
Response to Reviewer 2 Comments
Point 1: English expression and wording. I would reccommed having the English style and wording checked. I often stumbled across very uncommon expressions, which are not correct in my opinion. Also the grammar is frequently wrong.
Response 1: We have commissioned Proof-Reading-Service. com Ltd to modified the language before submission. Now, we have checked the manuscript clearly throughout the text again. We corrected some grammar errors and highlighted with yellow background as shown in the revised manuscript.
Point 2: The figure legends in general should be revised. The legends should indicate the type of assay and treatment conditions. They should further clearly indicate what is seen (e.g. colors in images).
Response 2: We have revised the figure legends as shown in revised manuscript with highlight yellow background.
Point 3: Selectivity of the drug. The authors state that the drug is more effective in cancer cells, however they show that the IC50 for one of the HCC lines and the hepatocyte cell line used is almost identical (1.10 vs. 1.13). The authors fail to explain this. Further, the IC50 only differs 2-fold in the other cell lines. The authors should elaborate on why they think this drug might be selective and further cell lines should be tested. Additionally, the apoptosis markers and p38 activation should be checked (at least in key experiments) with another HCC line, as well as a healthy cell line to strenghten the claim.
Response 3:
Thank you for reviewer’s very valuable comments. Resistomycin has a potential therapeutic window with IC50 of 0.25–1.10 µM for HCC lines cells and 1.13 μM for the hepatocyte cell line (HL-7702), 3.06 μM for the normal human kidney cells (293T), developing it into cancer therapeutics needs to be cautiously performed. Considering strong activity of resistomycin and toxity of most anti-cancer drug, as a promising candidate leading compound, it is worthy to further modify and develop resistomycin into anti-cancer drug.
Resistomycin has been screened as the natural potential inhibitor of Pellino-1 as shown in our previous work (Signal Transduction and Targeted Therapy (2020) 5:133). Considering the flat structure character of Resistomycin, we can not rule out the possibility that its effect on cancer cell survival and apoptosis may be compounded by its multi-action and multi-mechanisms.
Additionally, We employed another HCC line huh 7 cells (p38 expression) to perform the extra experiment according to reviewer 's comments as shown in the flowing. The IC50 of resistomycin for huh 7 cells is 0.35± 0.21μM. Dapi staining showed that resistomycin treatment for 24 h caused characteristic apoptotic morpholog-ical alterations in a dose-dependent manner, including nuclear condensation and apop-totic body formation. Western bloting show that p38 activation involve in resistomycin -induced huh 7 cells apoptosis.
Point 4: Confocal images are too small. I cannot see the effects described in the text.
Response 4: We have re-edited the images. The arrow indicated that the phenomenon of nuclear condensation and apoptotic body formation appeared. These images are made by fluorescence microscope (OLYMPUS CKX53).
Point 5: As the authors correctly pointed out that the ubiquitin ligase Pellino-1 is affected by the drug, it is hard to distinguish the contribution of p38 and Pellino-1 to the phenotype observed. The authos should try to clarify this by additional experiments (e.g. Pellino-1 inhibition or knock-down experments), or at least discuss this in more detail.
Response 5: Thank you for reviewer’s valuable comments. In our previous paper (Signal Transduction and Targeted Therapy (2020) 5:133), we made effort to identify potential inhibitors of Pellino-1. Among our library which contains 354 compounds, we found that resistomycin displayed a high affinity for GST-Pellino-1 (KD = 2.58 μM). Of course, as a flat structure natural product (similar to Aspirin, Metformin), it is possible that resistomycin has multitargets. Pellino-1 is a recent discovered E3 ubiquitin ligase. More and more reports are revealing Pellino-1’s novel effects and foundation. We have not evidence whether Pellino-1 involve in p38 pathway and furthermore related p38 and Pellino-1 experiments are warranted.
Minor issues
- line 45 "most studied" I doubt that. The context should be clarified
- line 62 "Pellino" is first mentioned here and should be explained or introduced to the reader
- Fig 1B. Normalization to 100% of Control should be visualized in the figure
- Several downstream effects of p38 activation or inhibition have been mentioned in the discussion. It would be benefitial to investigate some of these in the study.
Response to minor issues: we have made correction according to the Reviewer’s comments. We will investigate further in the follow-up study.

Reviewer 3 Report
The study is well conducted and the manuscript is well written. Certain minor mistakes need to check, such as the concentration unit umol/L or uM, and Res/Rec are not consistent throughout the text/Figures.
Another concern is: Results of Fig.4D and Fig.5B, have not been concordanced with the authors stated that pretreatment with SB203580 effectively reversed the function of resistomycin on apoptosis and cell cycle arrest when SB203580 functionally decreased the expression levels of phosphor-p38 and phosphor-Cdc25A. Perhaps, alternative pathways involve in Resistomycin induced apoptosis and cell cycle arrest.
Author Response
Response to Reviewer 3 Comments
Point 1: The study is well conducted and the manuscript is well written. Certain minor mistakes need to check, such as the concentration unit umol/L or uM, and Res/Rec are not consistent throughout the text/Figures.
Response 1: We have checked and corrected some minor mistakes throughout the text/Figures.
Point 2: Another concern is: Results of Fig.4D and Fig.5B, have not been concordanced with the authors stated that pretreatment with SB203580 effectively reversed the function of resistomycin on apoptosis and cell cycle arrest when SB203580 functionally decreased the expression levels of phosphor-p38 and phosphor-Cdc25A. Perhaps, alternative pathways involve in Resistomycin induced apoptosis and cell cycle arrest.
Response 2: We are sorry that we described inaccurately about the effect of SB203580 as shown in revised manuscript page1, line 21and manuscript page13, line 396. With or without SB203580, Resistomycin-induced apoptotic cells decreased from 23.15% to 9.80% and from 34.95% to 26.28% (Fig.4D), and the proportion of cells in the G2/M phase reduced from 25.39% to 19.22% and from 32.31% to 26.52% (Fig.5B) in the concentration of 0.25 and 0.5 μM, respectively. SB203580 could effectively impede resistomycin-induced apoptosis and cell cycle arrest. Resistomycin, as a planar-structured natural product, may have a multi-targeted action, we also agreed that there may be other pathways also involve the apoptosis and cell arrest processes.
